# Life-Long Experience with Male Mating Tactics Shapes Spatial Cognition and Coercion Evasion in Female Swordtails

**Philip S. Queller *, Elena R. M. Adams and Molly E. Cummings ***

Department of Integrative Biology, University of Texas, Austin, TX 78712, USA; elenaadams@utexas.edu
* Correspondence: pqueller@utexas.edu (P.S.Q.); mcummings@austin.utexas.edu (M.E.C.)

**Abstract:** Social experiences can shape adult behavior and cognition. Here, we use El Abra swordtails (*Xiphophorus nigrensis*) to assess how life-long experience with different male mating tactics shapes coercion evasion ability and female spatial cognition. We raised females from birth to adulthood in environments that varied by male mating tactic: coercers only, courtship displayers only, coercers and displayers together, mixed-strategists, and female only. In adulthood, we tested females' behavioral responses to a coercive male and spatial cognition in a maze. Females reared with only displayers were significantly worse at distancing themselves from the coercive male than females raised with coercers and displayers and females raised with only coercers. Females raised with a single mating tactic (either courtship display or coercion) exhibited significantly higher accuracy in the spatial maze than females from other rearing groups, and showed significant reduction in total errors (courtship display group) or latency to reward (coercion group) over successive trials. These more predictable environments (one tactic), and not the more complex environments (two tactics), showed evidence for spatial learning. The results are discussed in light of the existing literature on two components of environmental change (environmental predictability and the certainty with which cues predict the best behavioral response) and their effect on the development of cognitive abilities.

**Keywords:** spatial cognition; social competence; development; alternative reproductive tactics; predictability

**Key Contribution:** We found that the complexity and predictability of the social environment, specifically the number of male mating tactics present, determined adult female spatial learning performance. Females raised in more predictable sexual–social interactions (single tactics) developed greater spatial cognition than females raised with more (multiple tactics) or less (no males) social complexity.

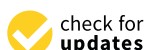



## 1. Introduction

Understanding how differences in cognition arise across individuals and species aligns questions from neuroscience to evolutionary biology, behavioral ecology, and psychology. Broadly, cognition refers to the neuronal processes that mediate the acquisition, processing, retention, and use of information [1,2]. Cognitive divergence across species, populations, and individuals is hypothesized to be driven by adaptation to specific ecological problems (see adaptive specialization hypothesis [3–6]), either through selection on genes or phenotypic plasticity. This hypothesis suggests that specific ecological and social pressures are associated with an investment in brain tissue and cognitive ability that facilitate animal responses to specific challenges.

There is extensive evidence that cognitive differences emerge in response to specific environmental challenges relating to foraging, predator evasion, and social contexts. For example, female cowbirds must successfully find and monitor host nests to parasitize, and they are associated with greater spatial cognition [7] and larger hippocampi [8] than males. Predation can shape cognition in poeciliid fish, leading to decreased performance in

associative learning and spatial tasks [9,10], but faster decision making and greater cognitive flexibility [10,11]. The social brain hypothesis was developed to explain primate brain evolution and argues that increased demands of group living (a proxy for social complexity) leads to a 'Machiavellian' intelligence that enables individuals to coordinate behavior within the context of the social group [12]. This hypothesis has been broadly extended into other animals to show that challenges of social living can shape cognitive evolution outside primates. For example, sociality in fish has been shown to be associated with numerous socio-cognitive traits [13], such as individual recognition [14] and the resolution of the prisoner's dilemma through cooperative predator inspection [15,16]. However, research with fish has revealed that individuals reared in isolation outperformed group-reared fish in a spatial task [17] and in inhibitory control [18], suggesting that not all cognitive functions scale with social complexity. Yet another aspect of the environment that can shape cognition is environmental predictability (or reliability). More predictable environments have been shown to favor memory and learning in birds [19] and drosophila [20,21], whereas unpredictable environments favor cognitive flexibility [19].

Developmental experiences can have a particularly strong effect on adult traits, and numerous studies have found that a variety of early life experiences can shape cognition in adults. In humans, early life stress has been routinely shown to have detrimental effects on adult cognition [22,23]; yet in mountain chickadees, elevated stress hormone levels are linked to superior spatial memory [24]. The enrichment of the physical environment increases both neurogenesis in the hippocampus and cognitive function in rats [25], and contributes to a larger brain volume in poeciliid fish [26]. In fish, researchers have begun to explore how early life experiences with different social group sizes can influence inhibitory control [18] and spatial learning [17], as well as how the stability of the social group (fission-fusion rates) can shape inhibitory control [18]. While social group size is a natural proxy for social complexity, few studies have investigated how variation in the type of social experiences during the entire life-span shape cognition in adulthood. Specifically, does the type of social interaction or the diversity of those types, rather than the absolute number of interactions, shape adult cognition?

Teleost fish from the family Poeciliidae offer a unique opportunity to manipulate specific socially salient experiences to observe their effect on adult female cognition. Poeciliids are live-bearing fish that are characterized by high levels of sexual conflict (i.e., conflict over mating rate due to differential reproductive investment) that results in frequent male coercion and female evasion. Coercion is costly to females due to reduced foraging efficiency and offspring fitness [27–31]. To evade coercion, females will actively avoid coercive males by increasing shoaling behavior [28,29] and spend more time in risky environments that males prefer to avoid [32]. In some poeciliid species, males will also court females. Here, we focus on one such species: the El Abra swordtail (*Xiphophorus nigrensis*). Males of this species exhibit one of three reproductive strategies that are determined by the copy number of the mc4r gene and can be inferred by body size [33–35]. Large males are ornamented and perform courtship displays to females; small males are drab and coerce females; and intermediate-sized males are modestly ornamented and perform a mixed-strategy of both courtship and coercion. Although not territorial, large males will aggressively guard females from other males [36]. Females prefer large males and avoid small males [37,38]. These multiple male mating strategies create a complex mating landscape that poses a variety of neurological and cognitive challenges to females [39,40], and allow us to ask how a specific fitness-related trait in females relates more broadly to spatial cognition.

Here, we reared females from birth to adulthood in rearing environments that varied in the type of male sexual behavior present: courtship display only (large males, D), coercion only (small males, C), coercion and display (small and large males, C+D), mixed-strategy (intermediate-sized males, M), and female only (F). These five environments vary along axes that are expected to influence cognitive development. Specifically, they vary in terms of complexity (highest in the C+D group with multiple phenotypes), stress (highest in the C group with more coercive males); and predictability (lowest in the M group

with unpredictable, mixed-strategy males). Previous work with a similar design found that experience with multiple mating tactics (a proxy for social complexity) leads to the development of high boldness and low aggression behavioral syndromes in females [41]. Here, we ask whether differences in sexual-social rearing environments influence the development of a female's coercion evasion abilities and spatial cognition.

We predict that rearing with the most stressful social environment (all coercive males) will enable those females to have a greater ability to avoid a coercive male. In addition, we predict that females raised with all coercive males will have greater performance in the spatial maze than other treatments, because heightened spatial cognition may aid the female avoidance of coercive males, and based on previous results with birds showing environmental stress favoring spatial memory [24]. Alternatively, if social complexity drives spatial cognition, we may expect to find that females raised with two mating tactics (C+D and M treatments) will have the greatest performance in the maze. Finally, if predictability favors learning, we expect M females to show the weakest evidence for learning, because females experience males that can unpredictably switch between display and coercion.

## 2. Materials and Methods

### 2.1. Social Rearing Environments

We raised female *Xiphophorus nigrensis* from birth to adulthood (1–1.5 years) in five socially controlled rearing treatments (Figure 1A–E), followed by cognitive-behavioral testing in early adulthood. The experimental females were first introduced into experimental aquaria (50 × 25 × 28 cm) as fry (<10 mm) from broods produced in semi-wild outdoor community tanks at UT's Brackenridge Field Laboratory where adult females used as models had experience with all male types. Individual broods were split across all treatments to control for genetic effects. The 43 experimental aquaria (five treatments with 7–9 replicate tanks each) were initially stocked with 10–12 fry of unknown sex along with two adult model females (or four in the female only group) and two adult males (except for D treatments, see below for more details). The fish were fed Cargill and TetraMin flakes once daily and were supplemented with brine shrimp periodically. The tanks were enriched with plastic plants, flowerpots, and gravel, and were illuminated with full spectrum aquarium lights. The experimental tanks were visually inspected three times a week to remove developing males. Developing males can be identified by a thickening of the anal fin as it develops into a gonopodium. To minimize the variation in social densities across treatments and replicates over time, fry would be added to or removed from replicate tanks to maintain similar densities (11–14 fish per tank). Adult male stimuli were also moved between replicate tanks within a treatment every three months to ensure that the developing females experienced multiple individual males of a given phenotype.

Our experimental females were reared in one of five sexual-social rearing treatments that varied by the type of male sexual behavior that the females experienced. The rearing treatments varied by the different types of stimulus males present: large courtship-displaying males only (D), small coercive males only (C), mixed-strategy intermediate-sized males only (M) that both courted and coerced females, a combination of small coercive and large displaying males (C+D), and females only (F). To control for adult density, each tank had four stimulus adults including two males and two adult model females (except for the D group, that had one adult male and two adult females, see below). This design differed slightly from a previous iteration of this experiment in which we had twice as many adult models (four males and four adult model females) in larger tanks with more juveniles [41]. We assume that the experimental females in the current study experience differences in male behavior across treatments as found in the previous study, when females raised in the D, M, and C+D sexual-social treatments experienced higher courtship than females raised in the C and F treatments (ANOVA: $F = 61.2$, $p < 0.001$); and females raised in the C, M, and C+D treatments experienced more coercion than females raised in the D and F treatments ANOVA: $F = 10.5$, $p < 0.001$; [41]).

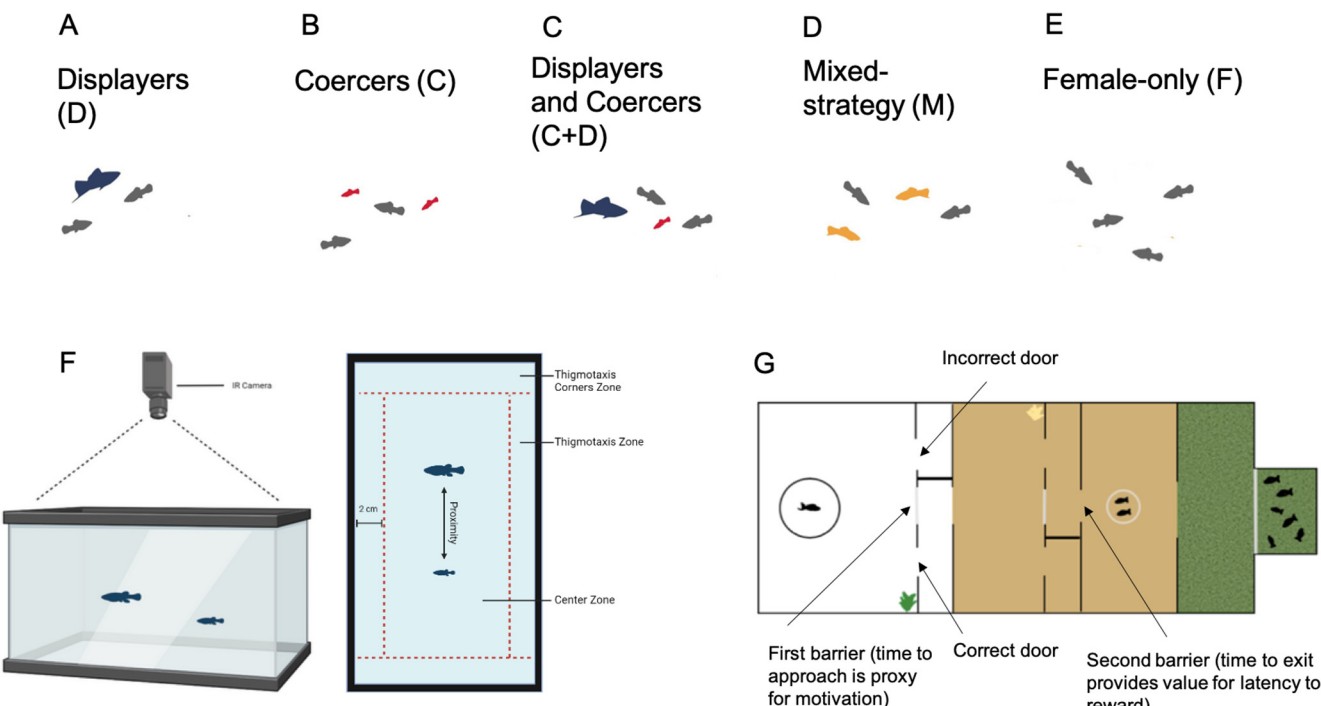

**Figure 1.** Schematic of the rearing environments (**A**–**E**), coercion evasion tank and scoring regions (**F**), and route learning apparatus (**G**). Females were raised with two adult model males and two adult model females, except for the D treatment that had only one adult male (to avoid male–male competition and ensure courtship display towards females). (**F**) The coercion evasion assay placed individual focal females with a free-swimming adult coercive male for 10 min. We used automated tracking software (Ethovision) to track the movement data of females including proximity to the coercive male, the amount of time spent in sheltered zones (thigmotaxis regions), and time spent in the exposed center region. (**G**) The route learning assay involved releasing a focal female from the habituation cylinder and measuring her motivation to engage in the task as measured by her latency to approach the first barrier (or component). Each of the two barriers had a transparent window down the center line that enabled focal females to view the social reward. After release, the female's spatial learning was assessed by her ability to navigate two barriers (or components) that had 'correct' and 'incorrect' doors that would allow passage through the maze to reach the social reward (two conspecific females plus a larger female shoal behind them). The females were tested across five trials, and we measured door choice accuracy, total errors made (number of incorrect door choices), and latency to reward across trials.

Model females were selected from the same community tanks as the fry, originally from semi-wild outdoor populations, and had experience in natural sexual-social conditions with all male mating strategies present. These females served as social learning models for the developing females. To minimize intrasexual aggression in the large, courting male treatment (D), we provided only a single male stimulus along with two adult models females in each replicate tank. The C+D group contained one small, coercive male and one large, displaying male. The M and C groups each had two males of their respective phenotype, and the F group had four adult model females. We selected large adult females (usually >30 mm) to serve as models in our treatment tanks. The adult model females were dorsally tagged with white elastomer markings to differentiate them from the developing experimental females. It takes approximately one year for *X. nigrensis* females to reach sexual maturity, and sexually mature females are distinguishable from immature females by a melanized brood patch on the ventral area near the gonopore. We maintained the experimental females in the treatment tanks for 15–18 months in order to characterize developmental effects from the combined influences of social learning (from observing

model females) and their own direct experiences as adults. As mate choice copying has been documented in other poeciliids, we expect social learning of mate preferences to strongly shape developing female preferences in adulthood. Fry were introduced to the rearing environments in April and May 2019 and tested as adults from June to August 2020.

### 2.2. Schedule

When the females reached adulthood, as indicated by the melanized brood patch, we tested them in a battery of cognition and behavior assays. The females were tested on one assay per day for nine days (see Supplemental Table S1 for assay order). Here, we report data on route learning (day 2) and coercion evasion (day 7), and data from other assays are published in [41] and will be published elsewhere.

### 2.3. Coercion Evasion

Prior to female coercion evasion testing, stimulus males (small coercive males) were gathered into two tanks and housed together for the duration of the testing period (from June to August) at densities of 4–6 males per tank. Thus, these coercive male stimuli were highly motivated to engage in sexual harassment during the testing trials given the single-sex housing conditions. Each individual coercion evasion trial was conducted with a new male on a given testing day, with the exception of only two trials in which the same male was used twice in one day. The same group of males was reused across different bouts/days of testing. We used males from the 23–25 mm size class for 55 trials, with 3 trials using males smaller than 23 mm.

The coercion evasion apparatus consisted of a 20 × 40 cm tank, with gray felt on the inside (Figure 1F). Two rectangular sheets of infrared (IR) transmittance plastic were placed inside the long sides of the tank, with the bottoms angled slightly inward to prevent the fish from swimming directly along the edges of the tank bottom, because the IR lights could not fully illuminate these regions. Two square pieces of blue foam material were placed along the short sides of the tank to prevent the fish from swimming behind the IR transmittance sheets. IR LED lights were placed underneath the tank and also wrapped around all four sides. No additional light sources were aimed at the tank, but the overhead fluorescent lights of the room were left on. The IR camera was mounted directly above the tank to record the trials. When the fish were in the tank, the top was covered with an RSCO 62 blue filter and a R3403 neutral density filter. The tank was filled with 10 cm of water, and 50% of the water was replaced before each new trial.

At the start of each trial, one focal female was retrieved from her holding tank, and immediately placed in a white PVC tube inside the coercion evasion tank to habituate for 5 min. After the 5 min habituation, the PVC cylinder was removed, and the focal female was allowed to swim freely for 10 min, recorded by the IR camera. A small stimulus male was then retrieved from the small male housing tank, and placed in the white PVC tube inside the coercion evasion tank—with the female still in the tank outside the PVC tube—to habituate for 5 min. After the 5 min habituation, the PVC tube was removed, and the male and focal female were allowed to swim freely about the tank together for 10 min, recorded by the IR camera. The focal female was then returned to her holding tank, and a drop of StressZyme was added to the water. The small male was returned to a new tank to differentiate which males had already been used in a trial that day. The above process was then repeated for each new focal female. Automated tracking software (Ethovision XT 15, Noldus, Wageningen, The Netherlands) was used to record each trial, in conjunction with Basler Pylon Viewer camera software.

### 2.4. Route Learning

We used a route learning apparatus adapted from [42]. The tank (70 × 29 cm) was filled with 13 cm of water and was evenly divided into three sections with two maze components (Figure 1G). Each maze component had a correct entry leading to the next section and an incorrect entry leading to a dead end. The correct doorways for each of the

maze components were marked in the tank with a synthetic plant near the outside of the doorway. As indicated in Figure 1G, the middle part of the first and second barriers was covered with a transparent mosquito net to allow test fish a clear view to the area containing the reward. The tank was designed to be less stressful for the fish as they approached the reward zone. The walls and floor of the start zone of the tank were covered in white felt and brightly illuminated, the middle section was lined with beige felt, and the reward zone was lined with dark green felt and gently illuminated by covering aquarium lights with blue filters. Gravel in the bottom of the dark green section provided a naturalized setting. In the third section, a small, transparent tube housed two females with brood patches as an initial social reward. We also included an adjacent tank with six females positioned flush against the glass of the reward zone as an additional social reward. During trial 1, the focal females were initially placed in a PVC cylinder in the start zone for a 5 min habituation period, and then released and allowed 10 min to reach the reward. The females that did not reach the reward were gently guided to the reward zone. After each trial, the females were constrained in the reward zone for 2 min to avoid potential aversive learning associated with consecutive netting. For trials 2–5, the females habituated for 2 min in a PVC tube in the start zone.

We evaluated motivation and learning performance within each of the five social rearing treatments. To evaluate motivation to engage with the route learning task, we quantified the latency of females to approach the first maze component (any region of the first wall). We measured three aspects of performance in the maze: door-choice accuracy, error reduction, and latency to reward. Because latency to reward began to increase in later trials (4 and 5), we excluded trials 4 and 5 from all learning performance analyses to minimize confounding factors (see Section 3 and Supplemental Figure S1). To calculate door-choice accuracy, we counted whether the fish made a correct or incorrect door choice at each maze component. We only counted a focal fish's first choice at each component (2 per trial); if the fish swam through a door it had previously entered but in the opposite direction of the reward, these subsequent choices were not counted towards their accuracy score. We excluded trial 1 from our analysis on accuracy because the fish had no prior experience with the apparatus and it was not expected to differ from chance. A fish's total accuracy score was calculated as the total number of correct choices out of the total number of choices across trials 2 and 3. To evaluate learning based on the reduction in errors across successive trials, we calculated the total number of errors (e.g., total number of incorrect door entries made) in each trial and determined whether each rearing treatment exhibited a significant negative slope in total errors across trials 1 to 3. For this metric, if the focal fish made multiple choices at the same door, we still included those incorrect choices as part of the error calculation, but we only included choices made when swimming in the direction of the reward. Our third learning metric was a reduction in latency to reward across successive trials. We calculated latency to reward (time to enter the reward zone–time to approach first barrier) and determined whether each rearing treatment exhibited a significant negative slope in latency to reward across trials 1 to 3.

*2.5. Statistics*

We used R version 4.1.2. in all analyses. Shapiro–Wilkstests were used to assess normality of variables and model residuals. We used Wilcoxon rank sum tests for nonparametric pairwise comparisons. For comparisons with more than two groups, we used ANOVA for parametric variables and Kruskal–Wallis tests for nonparametric variables. We used Tukey post hoc tests for ANOVAs and Dunn post hoc tests for Kruskal–Wallis tests.

To assess differences in coercion evasion across treatments, we compiled measures from the coercion evasion assay collected using Ethovision tracking into a principal component analysis (PCA). We compared PC1, PC2, and PC3 (based on automated PC selection from [43]) across treatments using an ANOVA.

To assess differences in motivation to engage with the spatial maze, we compared the time to reach the first barrier (proxy for motivation) between treatments using a

Kruskal–Wallis test. To test for learning in the spatial maze, we assessed each treatment for 3 different learning metrics: higher than chance door-choice accuracy, significant reduction in number of errors, and significant reduction in latency to the reward. To test for choice accuracy in the maze, we used Wilcoxon tests to see if treatments differed significantly from chance (50%), with groups showing significantly higher than 50% accuracy as evidence for learning. To test for a significant reduction in number of errors, we ran generalized linear mixed-models (GLMMs) with a gamma distribution and log link function, with total errors + 1 as the response variable, trial as a fixed effect, and replicate tank as a random effect (glmer() from lme4 package in R). We chose GLMMs with gamma distribution for this metric because there were a disproportionately large amount of zeros and a large positive skew in the total errors variable. To test for a significant reduction in latency to reward, we ran linear mixed models (LMMs) with log transformed latency to reward as the response variable, trial number as a fixed effect, and replicate tank as a random effect (lmer() from lme4 package in R). A significant effect of trial with a negative slope shows the fish significantly reducing their number of errors or latency to reward over consecutive trials and provides additional evidence of learning.

## 3. Results

In total, we tested 56 and 55 experimental females that had spent their entire lives (12–15 months) in one of the five social rearing treatments in the coercion evasion and route learning maze, respectively. Specifically, we tested 9 F females, 10 M females, 9 C+D females, 13 D females, and 15 C females in the coercion evasion assay. The same females were also tested in the route learning assay, with the addition of one C+D female ($n = 10$ total) and two fewer F females ($n = 7$ total).

### 3.1. Coercion Evasion

To evaluate female coercion evasive behavior, we computed a PCA using 14 movement variables collected during the 10 min coercion evasion assay and tracked with Ethovision software (using the SIM module). We extracted the first three PCs following automated methods found in [43]. PC1 accounted for 23.8% of the variation and was driven largely by positive loadings that related to cumulative time in close proximity to male (1 cm and 2 cm) and frequency of body contacts, with negative loadings driven by the maximum distance between female subject and male (Figure 2A). Hence, PC1 is an indicator of effective coercion evasion techniques, with higher values representing a failure to avoid the coercive male while lower or negative values representing the successful avoidance of the coercive male. We found that PC1 scores significantly differed across treatments (ANOVA $F = 2.74$, df = 4, $p = 0.039$, Figure 2C). A Tukey post hoc test showed that this is driven by C+D females having significantly lower PC1 scores than D females ($p = 0.045$), and C females having marginally significant lower PC1 scores than D females ($p = 0.058$). This result indicates that females from the treatment where only displaying males were present spend more time in close proximity to the coercive male than females from treatments where coercive males are present (C and C+D groups). There was no difference between groups in PC2 (20.6% of variation) scores ($F = 0.79$, df = 4, $p = 0.53$) or PC3 (15% of variation) scores ($F = 0.67$, df = 4, $p = 0.61$).

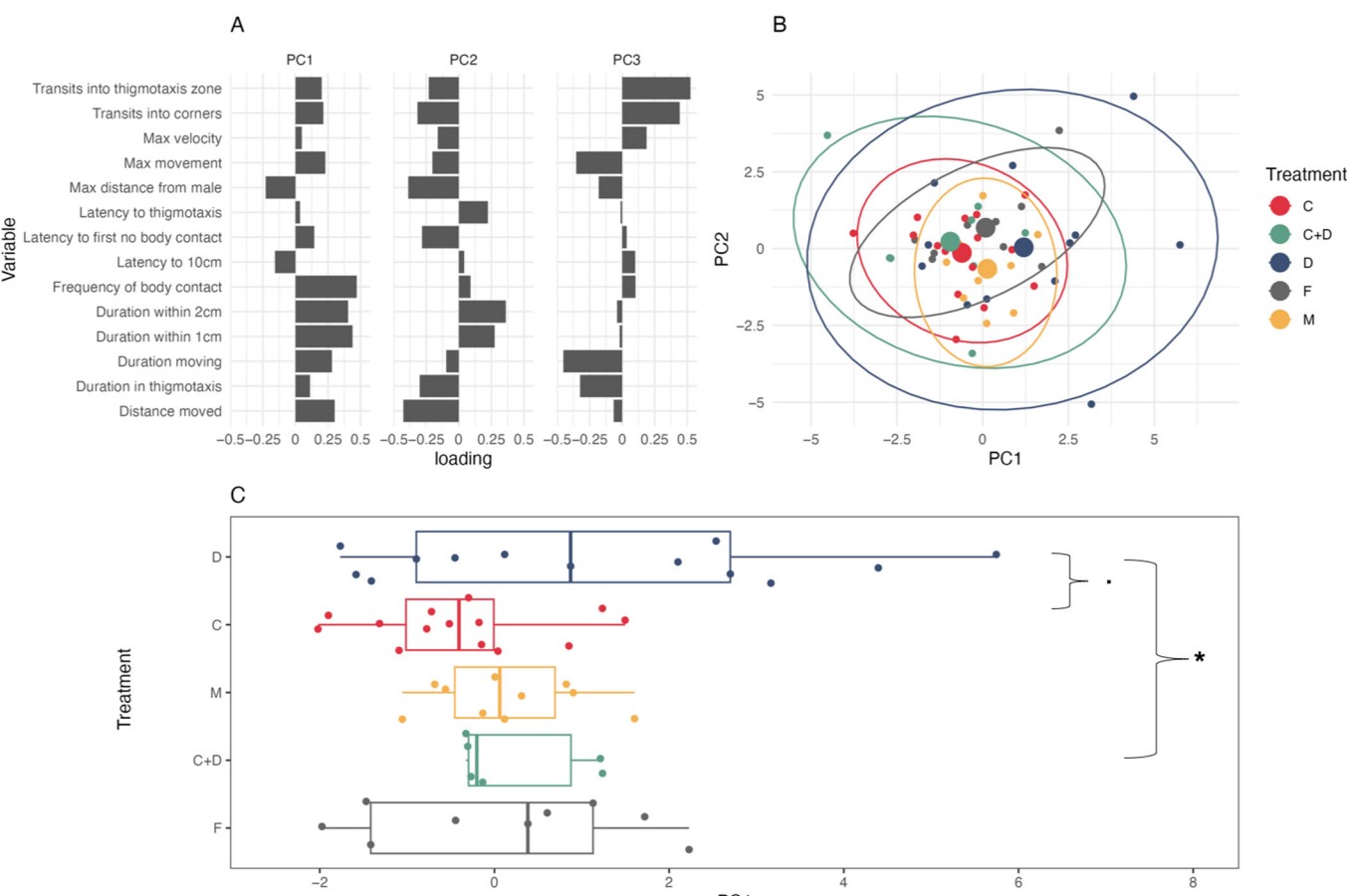

**Figure 2.** Coercion evasion behavior differences across females from different sexual-social rearing treatments. (**A**) Eleven behavior variable loadings for PC1, PC2 and PC3 of the coercion evasion PCA. PC1 loadings are largely comprised of behaviors that relate to a female's distance from a coercive male. (**B**) PCA plot of PC1 and PC2 scores by five sexual-social rearing treatments. Points are colored by treatment (C: red; D: blue; C+D: green; M: gold; and F: grey), with centroids representing treatment means and ellipses representing 95% confidence intervals. PC1 explains 23.7%, PC2 explains 20.6%, and PC3 explains 15.4% of the variation. (**C**) Significant difference in PC1 scores across treatments (ANOVA $F = 2.74$, $p = 0.039$), with Tukey post hoc test showing that D females had significantly higher PC1 scores than the C+D females ($p = 0.045$) and nearly significantly higher PC1 scores than C females ($p = 0.058$). The boxplot shows medians (vertical lines), interquartile ranges (boxes), and whiskers (horizontal lines), with individual data points colored by treatment. The asterisk and dot indicate post hoc significant and marginally significant differences, respectively.

*3.2. Route Learning*

We tested 55 females from different rearing environments and assessed their motivation to engage in the learning task followed by three learning performance metrics: choice accuracy, reduction in error rate over multiple trials, and reduction in latency to reward over multiple trials. First, we looked for differences in motivation. We found that the average motivation measured across five trials differed across treatments (Kruskal–Wallis: $X^2 = 21.24$, df = 4, $p = 0.0003$, with post hoc tests showing F females being less motivated than M ($Z = -3.577$, $p = 0.003$) and C+D females ($Z = -4.28$, $p = 0.0002$). There was also a trend for F females to be less motivated than D females ($Z = -2.70$, $p = 0.06$), and for C+D females to be less motivated ($Z = 2.48$, $p = 0.09$) than the C females (Figure 3). To control for this, we subtracted motivation values from latency to reward such that differences in willingness to engage with the apparatus were not included in this measure of performance (e.g., the start time for latency to reward begins once the female engages with component 1 of the maze). We also observed that latency to reward began to increase during trials 4 and

5 (Supplemental Figure S1) and excluded these trials from further analysis, because this increase is likely due to habituation to the tank or the stress of consecutive netting, as found in other studies in poeciliids [42]. Examining motivation differences across social rearing treatments for trials 1 to 3 (removing trials 4 and 5) reduced the highly significant difference to a marginally significant one that maintained the same pattern of F group females being the least motivated (Kruskal–Wallis: $X^2$ = 9.45, df = 4, $p$ = 0.051).

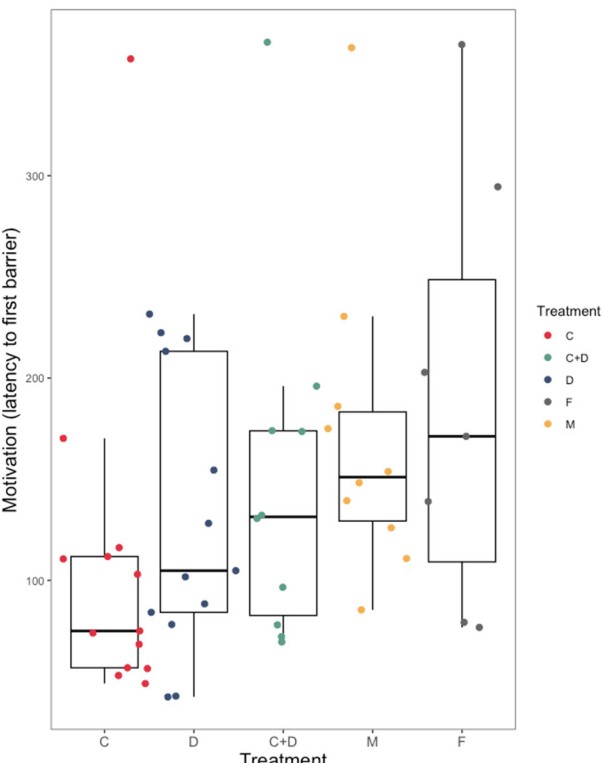

**Figure 3.** Differences in motivation between treatments across five trials ($p$ = 0.0003). Post hoc tests show that F females are less motivated than M ($p$ = 0.003) and C+D females ($p$ = 0.0002). The boxplot shows medians (horizontal lines), interquartile ranges (boxes), and whiskers (vertical lines), with individual data points colored by treatment. Note that lower values signify faster latency to the first barrier and represent higher motivation.

We next asked how each social treatment performed across three learning metrics. We found that females from predictable male mating environments (one mating tactic, including both the C and the D groups) performed significantly better across our three learning metrics than all other groups. In terms of accuracy in door choice across trials 2 and 3, only the C and D groups selected the correct door as their first choice across both maze components (door choice accuracy), with a significantly better than 50% chance: C ($t$ = 2.92, df = 12, $p$ = 0.006), D ($t$ = 2.31, df = 12, $p$ = 0.020), C+D ($V$ = 5, df = 9, $p$ = 0.58), M ($t$ = −0.32, df = 9, $p$ = 0.62), and F ($t$ = 0, df = 6, $p$ = 0.50); see Figure 4A. A reduction in total errors across trials 1 to 3 was only significant in D females (estimate + SE = −0.28 ± 0.11, $t$ = −2.53, $p$ = 0.01, residuals: $W$ = 0.95, $p$ = 0.09), but not in other groups (C: estimate = −0.20 ± 0.17, $t$ = −1.20, $p$ = 0.23, residuals: $W$ = 0.95, $p$ = 0.29; C+D: estimate = −0.006 ± 0.23, $t$ = 0.002, $p$ = 0.99, residuals: $W$ = 0.94, $p$ = 0.13; M: estimate = −0.03 ± 0.23, $t$ = −0.14, $p$ = 0.89, residuals: $W$ = 0.94, $p$ = 0.18; F: estimate = −0.10 ± 0.25, $t$ = −0.40, $p$ = 0.69, residuals: $W$ = 0.96, $p$ = 0.57) (Figure 4B). Finally, only females from the C group exhibited a significant reduction in latency to reward across trials 1 to 3: (C: estimate = −0.58 ± 0.24, $t$ = −2.41, $p$ = 0.03, residuals: $W$ = 0.93, $p$ = 0.10; D: estimate = −0.26 ± 0.17, $t$ = −1.52, $p$ = 0.14, residuals: $W$ = 0.98, $p$ = 0.74; C+D: estimate = −0.09 ± 0.26, $t$ = −0.36, $p$ = 0.72, residuals: $W$ = 0.95, $p$ = 0.22; M: estimate = −0.40 ± 0.26, $t$ = −1.54, $p$ = 0.14, residuals: $W$ = 0.97,

$p = 0.80$; F: estimate $= -0.16 \pm 0.35$, $t = -0.46$, $p = 0.65$, residuals: $W = 0.94$, $p = 0.23$) (Figure 4C).

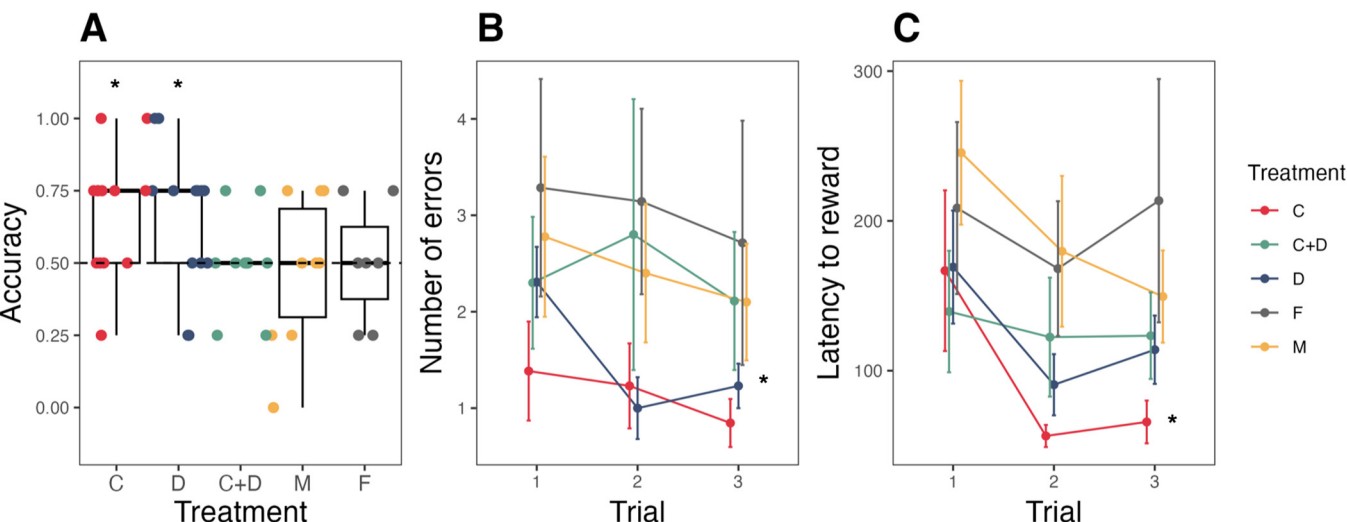

**Figure 4.** Females from different sexual-social rearing environments differ in route learning maze performance. (**A**) Females from the coercion only ($p = 0.006$) and courtship display only ($p = 0.02$) treatments exhibited spatial route accuracy that was significantly higher than chance (0.5), while those of other treatments did not. The boxplot shows medians (horizontal lines), interquartile ranges (boxes), and whiskers (vertical lines), with individual data points colored by treatment. The dotted line denotes 50% accuracy expected by chance, and the asterisk signifies significant difference from chance for each group. (**B**) Females from the display only group showed a significant reduction in the number of errors over trials 1 to 3 ($p = 0.01$), while those from other treatments did not. (**C**) Females from the coercion only group significantly decreased their latency to the reward (seconds) over trials 1 to 3 ($p = 0.03$), while those of other groups did not. In (**B**,**C**), the circles represent mean values for each treatment, and the vertical lines represent standard error. The asterisks in (**B**,**C**) represent a statistically significant effect of the trial for the marked group.

## 4. Discussion

In this study, we successfully created discrete sexual-social environments that females experienced for the entirety of their developmental period (from parturition to adulthood) by manipulating exposure to different combinations of male alternative reproductive phenotypes. These different sexual-social experiences shaped how females (un)successfully avoided coercive males, as well as how well they learned a spatial maze. Interestingly, we found that simple sexual-social environments in which females experienced only a single mating tactic (either only coercion or only courtship) resulted in better spatial learning than dual mating tactic environments or single-sex rearing environments.

### 4.1. Coercion Evasion

We found that females reared with only courtship-displaying males were more likely to spend greater time in proximity to the coercive male (higher PC1 scores, Figure 2) than females reared with both coercive and displaying males or females reared exclusively with coercive males (Figure 2). These results support our hypothesis that life-long experience with coercion improves how females respond to harassment during adulthood. The greater experience with coercion in the C and C+D environments appears to have given these females an advantage in developing effective behavioral responses (i.e., minimizing the time spent in proximity to a harassing male) relative to females who were only familiar with courting male types. This result is expected if experience with coercion shapes a female's evasive response to sexual harassment, as the D females were completely naïve to coercive behavior. One result that is unexpected is that coercive only females were not

significantly better at avoiding time spent near coercive males than females who were completely unfamiliar with males (F females; Figure 2C). We suspect that females from the male-naïve group may have exhibited neophobia to the novel entity (and hence exhibited avoidance to the novel male stimuli rather than to the behavior he was attempting).

Avoiding sexual harassment from males is an important fitness-related behavior for females, because coercion reduces female choice, increases the risk of injury and infection, and is costly enough that the females of some poeciliid species will venture into areas with increased predation to deter males [32]. Our results suggest that females learn 'avoidance' throughout their lifespan, and that more experience with harassment leads to greater abilities to avoid harassment. They also suggest a parallel to how female *X. nigrensis* swordtails appear to learn mate preferences. The strength of female preference for large, courting *X. nigrensis* male phenotypes increases as females age [44], and the results we show here suggest that the female avoidance of small, coercive males also gets stronger with greater experience.

Social-rearing experience has been shown to shape adult social behavior in a number of species. In guppies, males raised with adult males and females performed longer courtship displays towards females than males raised without the opportunity to learn from adult males [45]. In cooperatively breeding cichlids (*Neolamprologus pulcher*), rearing with a breeding pair (vs. no breeding pair) increases the number of aggressive and submissive displays when focal fish win or lose contests, respectively, as well as increasing subordinate behavior in a novel context (submitting to a new breeding pair in an attempt to join the social group) [46]. These behavioral changes in response to social rearing are also accompanied by differential neurogenomic response [47] and brain morphology [48]. Furthermore, social isolation impairs cooperative predator inspection behavior in cichlids (*Pelvicachromis taeniatus* [49]). These studies can be broadly linked by the concept of social competence [50,51], which refers to the observation of individual variation in the speed, accuracy, and quality of social-decision making. In this study, females raised with displaying males were less competent in avoiding a coercive male than females raised with coercive males, indicating these females have a lower social competence as a result of their life experience.

*4.2. Spatial Cognition*

We predicted that females from the coercion only environment would exhibit better spatial cognition than all other rearing environments, due to the selective pressure to avoid sexual harassment and previous research with birds showing a positive effect of stress on spatial memory [24]. However, we did not find exclusive support for that. Instead, we found that females reared in both of the highly predictable, simple sexual-social environments (coercive only and courtship display only) showed evidence of learning in the spatial maze (Figure 4A,B), while females from complex sexual-social environments (dual tactic environments) or overly simple environments (female only environments) did not. The higher accuracy among coercion only and courtship only females, coupled with a significant reduction in error rate for females reared with only displaying males, and a reduced latency to reward among females reared exclusively with coercive males, indicates that females from simple sexual-social environments learned while females from more complex environments did not. While this pattern does not fully rule out our hypothesis that experience with increased coercion will facilitate an increased investment in spatial cognition, it does suggest that something beyond coercion experience is driving cognitive performance.

Social complexity is thought to be a major driver of cognitive evolution and development [52,53], so why is it that females reared in more simple environments have greater spatial cognition? Contrary to a common assumption that complexity enhances cognition, there have been a surprising number of experiments showing otherwise. For example, guppies reared in either isolation or stable social groups displayed greater inhibitory control compared to group-reared or unstable social groups, respectively [18], and guppies

reared in low social densities show an increased ability to locate a food item in a spatial maze relative to subjects reared in high social densities [54]. Similar experiments with the cichlid fish (*Pelvicachromis taeniatus*) revealed that fish reared in isolation had better spatial cognition in a maze than fish reared in a social group [17]. Nonetheless, there are examples where social isolation led to the impairment of spatial learning [55]. In our experiment, we did not vary the social environment by the quantity of conspecifics present, but rather by the number and type of male behaviors the female subjects experience. These results offer some interesting new insights into the cognitive benefits and costs of simple vs. complex social environments.

Beyond varying along a social complexity axis, our social manipulation experiment contained experimental environments that vary in terms of the predictability of social interactions as well as the certainty with which these interactions predict the best behavioral response [20,21]. In terms of environmental predictability, social treatments varied from the most predictable (F females, as social interactions would be 100% with females) to the least predictable (M females, where females encounter other females but also highly unpredictable, mixed male strategists), and some treatments that fall in-between this spectrum. In general, the predictability of the social environment varied across our social rearing treatments as a continuum (from low to high predictability: M < C+D < D, C < F; see Figure 5). In terms of the certainty with which the interactions predict the best behavioral response in females (e.g., the certainty of best action to maximize fitness when met with a stimulus [20,21]), social treatments varied from high certainty in the F treatment to low certainty in the C+D treatment. We contend that females in the F group had the highest certainty as they had a single adult phenotype to interact with (other females) and no mating decisions, and therefore the certainty with which the interaction with other females would predict the best behavioral response is high, and in this case, likely determined by foraging decisions. Meanwhile, females reared in environments with adult females plus a single male phenotype (coercive, courtship displayer, or mixed-strategist) had lower certainty because their response towards these monotypic males will vary based on a female's reproductive state (to mate or not to mate). Finally, the certainty with which cues predict the best behavioral response may be lowest in the social environment with three adult phenotypes (adult females, coercive males, and displaying males found in the C+D environment), as the best response will vary with the reproductive state of the females and with the type of the interacting partner (coercive male or courting male or another adult female). Different combinations of these two components of variation (predictability and certainty of best response) are expected to favor or disfavor learning (see Figure 5, a modification of [20,21]).

Here, we posit that the combination of high environmental predictability with intermediate certainty of the best response favored the learning performance observed in the coercion only and the courtship only rearing environments. Pioneering research by Dunlap and Stephens with *Drosophila* (both modelling and empirical) has indicated that these two components of variation contribute to learning in a non-parallel way [20,21]. When experience with environmental stimuli has high reliability (highly predictable environment), learning is favored only when there is some uncertainty of the best response to maximize fitness (see Figure 5). Conversely, when there is low reliability of experience (unpredictable environment) but complete certainty of the best response to maximize fitness, then there is no need for learning to develop. In our experiment, the M group females were raised with the lowest reliability of experience as mixed-strategy males represent unpredictable social stimuli, and this may have contributed to their poorer spatial learning performance relative to the more predictable courtship only and coercion only rearing groups. Meanwhile, the rearing group with the highest predictability (female only) also represented the social environment with the greatest certainty with which cues predict the best behavioral response, and this combination is selectively neutral with regard to learning, similar to an environment that is represented by intermediate levels of both predictability and certainty (e.g., C+D females; see Figure 5). Predictable environments (e.g., high reliability of experience)

have been shown to favor learning in drosophila [20,21] and guppies [56]. For example, guppies exposed to environments with predictable food availability developed enhanced learning abilities but had lower cognitive flexibility than fish reared with unpredictable food resources [56]. While it is usually the predictability of the physical environment (e.g., food or oviposition substrate) that has been examined for its effect on learning and memory, it is interesting to consider the parallel impact that the social environment may produce.

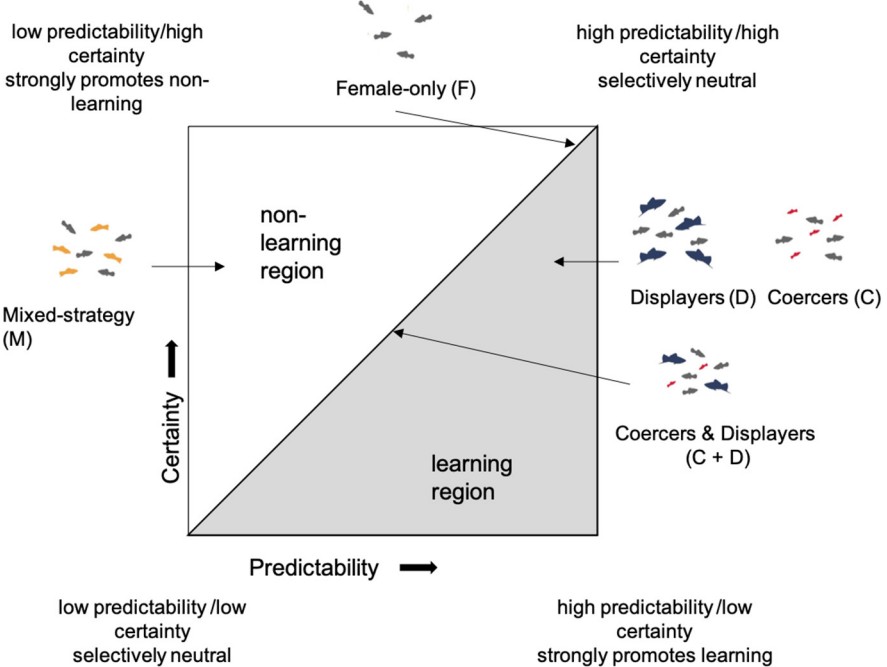

**Figure 5.** Visualization of the effects of the certainty with which cues predict the best behavioral response (y axis) and environmental predictability (x axis) on learning (adapted from [20,21]). Females raised with only coercers (C) or only courtship displayers (D) have high levels of predictability and moderate levels of certainty, placing them in the region where learning is promoted. Mixed-strategy-exposed females (M) have low predictability and moderate certainty, which places them in a non-learning region of the model. Females reared with no exposure to males (F) have high predictability and high certainty, and should be found in the region where learning is selectively neutral. Similarly, females reared with both coercers and displayers have intermediate levels of both predictability and certainty, which also places them in a region of the model that is selectively neutral for learning and non-learning.

Our previous research using sexual-social rearing manipulations with *X. nigrensis* found that females reared in simple sexual-social environments (courtship only or coercion only) were less bold than females from complex environments with two mating tactics [41]. In comparison to the present study, we find that these simple and predictable sexual-social rearing environments increase spatial cognition relative to the more complex and less predictable social environments. Interestingly, boldness and cognitive abilities often positively correlate across taxa [57–59]; however, this pattern is not universal (see [60,61]). In the context of our sexual-social rearing environment studies, life-long experience with predictably simple social interactions increased spatial learning but decreased boldness. This suggests that the relationship between boldness and cognition may vary with environmental conditions and experience. More complex social environments may have greater levels of unpredictability that favor less investment in learning because of how quickly learned information may change, while boldness may be favored so that females can be aware of immediate conditions. Correlations between personality traits like boldness and cognition are often considered through multiple conceptual frameworks, including coping styles [62] and pace-of-life syndromes [63]. We emphasize the importance to consider

long-term social experiences in the context of these frameworks and their ability to predict the relationship between personality traits like boldness and cognition.

## 5. Conclusions

Here, we show that different life-long experiences with courtship and coercion shape coercion evasion behaviors and spatial cognition in *X. nigrensis* females. Females reared exclusively with displaying males until adulthood showed a worse coercion evasion response relative to females raised exclusively with coercive males or with a mixture of displaying and coercive males. While these single tactic environments resulted in the development of different coercion evasion responses, they converged on showing the highest levels of spatial learning performance relative to environments with two mating tactics or none at all (female only). We suggest that spatial learning performance increased in these single mating tactic environments due to an optimal combination of high predictability and incomplete certainty that favors investment in learning. Given that research in both birds [19] and fishes [10,56] reveals a trade-off between learning and cognitive flexibility, it is conceivable that females reared in the more complex sexual-social environments with higher levels of unpredictability (our dual mating tactic environments) may favor cognitive flexibility—a supposition that has yet to be tested. Future work should investigate other cognitive domains to assess cognitive trade-offs as a function of rearing experience, as well as comparative studies across other fish and species with mating system diversity to understand if similar patterns emerge over evolutionary time.

**Supplementary Materials:** The following supporting information can be downloaded at: https://www.mdpi.com/article/10.3390/fishes8110562/s1, Figure S1: Results from the route-learning maze including all 5 trials. Table S1: Schedule of assays including assays published elsewhere (aggresion, sociability, scototaxis in [41]) and assays to be published in subsequent articles.

**Author Contributions:** Conceptualization, P.S.Q., E.R.M.A. and M.E.C.; methodology, P.S.Q., E.R.M.A. and M.E.C.; software, E.R.M.A.; validation, P.S.Q., E.R.M.A. and M.E.C.; formal analysis, P.S.Q.; investigation, P.S.Q. and E.R.M.A.; resources, P.S.Q. and M.E.C.; data curation, P.S.Q.; writing—original draft preparation, P.S.Q.; writing—review and editing, P.S.Q., E.R.M.A. and M.E.C.; visualization, P.S.Q. and E.R.M.A.; supervision, M.E.C.; project administration, M.E.C.; funding acquisition, M.E.C. All authors have read and agreed to the published version of the manuscript.

**Funding:** This research was funded by the National Science Foundation (NSF) grant number IOS-1911826 awarded to Molly Cummings.

**Institutional Review Board Statement:** This experiment was approved by The University of Texas at Austin's Institutional Animal Care and Use Committee (IACUC) protocol number AUP-2022-00224.

**Data Availability Statement:** Data will be made available in a timely manner by the authors upon request.

**Acknowledgments:** We thank the staff at the University of Texas' Brackenridge Field Laboratories for care and maintenance of our fish populations. We thank members of the Cummings lab for their feedback on this manuscript and the reviewers for their insightful comments that improved the manuscript. Lastly, we thank the NSF for funding this research and to the College of Natural Sciences Summer TIDES fellowship to ERMA.

**Conflicts of Interest:** The authors declare no conflict of interest.

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
