# Peer review of "Life-Long Experience with Male Mating Tactics Shapes Spatial Cognition and Coercion Evasion in Female Swordtails"

_fishes, doi:10.3390/fishes8110562_

Round 1
Reviewer 1 Report (Previous Reviewer 1)
Comments and Suggestions for Authors
I read the revised version of this study and I think the authors made several interesting improvements. I do not have further suggestions. I hope to see this interesting manuscript published soon.
Author Response
Thank you for your helpful suggestions with our manuscript.
Reviewer 2 Report (Previous Reviewer 2)
Comments and Suggestions for Authors
The authors have responded adequately to all my concerns. I recommend publication of this manuscript.
Author Response
Thank you for your helpful suggestions with our manuscript.
This manuscript is a resubmission of an earlier submission. The following is a list of the peer review reports and author responses from that submission.
Round 1
Reviewer 1 Report
Comments and Suggestions for Authors
This study is very interesting and timing, dealing with a previously not investigated form of cognitive plasticity in fish. Overall, everything is as it should be and I would be happy to recommend publication of this study.
Please see below my detailed comments that may help improving the presentation of the study. I would probably say that the most important thing is fixing the interpretation of the time variable.
Abstract
I think that the introductory part of the abstract (the first sentence) is not sufficient. It seems to be that two aspects are investigated here, the cognitive plasticity and the ability to avoid the males. Probably the first sentence of two should reflect both these aspects. The abstract will be also more catching if it reports the hypothesis underlying the study an interpretation for the findings (stress, adaptation etc., please see also my comment on this point below).
Introduction
L30-35 Here the text is good but might be interpreted in different ways. I suggest to put it down clearly: adaptive cognitive differences can arise both as the product of genetic adaptation (i.e., selection on alleles/genes) or via phenotypic plasticity.
L48 I apologise for suggesting one of my own papers, but I think it is very relevant here.
We found that early experience with different types of social environment (large vs small, simple vs complex) altered cognitive function in guppies. I don’t recall similar studies in fish, so probably it is worth mentioning this one in introduction to justify your idea and see if the results somehow align or diverge from your results, therefore deepening the discussion.
Lucon-Xiccato, T., Montalbano, G., Reddon, A. R., & Bertolucci, C. (2022). Social environment affects inhibitory control via developmental plasticity in a fish. Animal Behaviour, 183, 69-76.
L56 Please check if these lines need also adjustments given the paper of above.
L83 I may be too picky, but I am not sure that the exposure to different mating strategies is only a ‘social’ treatment. Maybe it is ‘reproductive environment complexity’? If the authors disentangle this issue, then they can try to improve the title. I would see more precise (and more catching) something like “Early-life experience with male mating tactics shapes….”.
Methods
L103 Very nice number of replicates!
L125 Maybe habituation cylinder?
L131 How long was the real exposure to the male behaviour? I assume that until the females were sexually mature, the males were not courting them.
L158 Can 15-18 months can still be considered ‘early-life experience’ in this species? In many other poecilids it is close to the typical life span in the nature.
L200 maybe about can be removed?
L247 I would probably switch the order of the two tests, presenting first the statistical methods of the coercion experiment, in line with methods and results section.
L254 If I understand correctly, a decrease in the time to solve the task can be also due to other factors. For instance, the fish could be more and more habituated to the apparatus and progressively increase the swimming speed. If in this experiment it is not possible to exclude these effects, please do not referee to changes in time as learning.
Results
L286 Was the eigen value greater than 1? I don’t expect so. In case it is below 1, an analysis on the raw trait data may be more informative because the PC has probably lost a lot of behavioural variance.
L313 That is what I meant above. Probably the time measure is affected by factors other than learning. Maybe the time and the motivation better represent the approach to the task, or the style to tackle the task, rather than the learning performance. Please consider changing the text accordingly, I think it is quite important do don’t misinterpret this time effect. The main effect due to the route learning and the main result of the study should not be altered by this change.
Discussion
L405 Can it be that there are other factors involved in the treatment? Maybe the males with a specific mating tactic have also other behavioural features. See for instance:
Kelley, J. L., Phillips, S. C., & Evans, J. P. (2013). Individual consistency in exploratory behaviour and mating tactics in male guppies. Naturwissenschaften, 100, 965-974.
L471 Probably beside the paper suggested in introduction, these two can help expand the role of social environment in the present experiment (although they are not studies in fish):
Ashton, B. J., Ridley, A. R., Edwards, E. K., & Thornton, A. (2018). Cognitive performance is linked to group size and affects fitness in Australian magpies. Nature, 554(7692), 364-367.
Johnson-Ulrich, L., & Holekamp, K. E. (2020). Group size and social rank predict inhibitory control in spotted hyaenas. Animal Behaviour, 160, 157-168.
L491 I would check if this idea aligns with our most recent paper. It is a test on the effect of predictability on various cognitive functions in a poecilid.
Lucon-Xiccato, T., Montalbano, G., & Bertolucci, C. (2023). Adaptive phenotypic plasticity induces individual variability along a cognitive trade-off. Proceedings of the Royal Society B, 290(2001), 20230350.
L506 I think that domain specificity was demonstrated earlier here (at least in fish):
Montalbano, G., Bertolucci, C., & Lucon-Xiccato, T. (2022). Cognitive phenotypic plasticity: environmental enrichment affects learning but not executive functions in a teleost fish, Poecilia reticulata. Biology, 11(1), 64.
My compliments for this interesting study!
Author Response
This study is very interesting and timing, dealing with a previously not investigated form of cognitive plasticity in fish. Overall, everything is as it should be and I would be happy to recommend publication of this study.
Please see below my detailed comments that may help improving the presentation of the study. I would probably say that the most important thing is fixing the interpretation of the time variable.
Thank you for your valuable feedback!
Abstract
I think that the introductory part of the abstract (the first sentence) is not sufficient. It seems to be that two aspects are investigated here, the cognitive plasticity and the ability to avoid the males. Probably the first sentence of two should reflect both these aspects. The abstract will be also more catching if it reports the hypothesis underlying the study an interpretation for the findings (stress, adaptation etc., please see also my comment on this point below).
Good point. We have now added both elements of our study into the first sentence of the abstract. We have also added reference to the hypothesis our data supports at the end of our abstract.
Introduction
L30-35 Here the text is good but might be interpreted in different ways. I suggest to put it down clearly: adaptive cognitive differences can arise both as the product of genetic adaptation (i.e., selection on alleles/genes) or via phenotypic plasticity.
Agreed. This is now included (see lines 34-35).
L48 I apologise for suggesting one of my own papers, but I think it is very relevant here.
We found that early experience with different types of social environment (large vs small, simple vs complex) altered cognitive function in guppies. I don’t recall similar studies in fish, so probably it is worth mentioning this one in introduction to justify your idea and see if the results somehow align or diverge from your results, therefore deepening the discussion.
Lucon-Xiccato, T., Montalbano, G., Reddon, A. R., & Bertolucci, C. (2022). Social environment affects inhibitory control via developmental plasticity in a fish. Animal Behaviour, 183, 69-76.
This is an extremely interesting study that we somehow missed— thank you kindly for directing our attention to it. This research inspired us to re-pitch our introduction and we now include this reference in the introduction (lines 56 and 65-67) as well as in the Discussion (line 509).
L56 Please check if these lines need also adjustments given the paper of above.
Completely rewritten given the above.
L83 I may be too picky, but I am not sure that the exposure to different mating strategies is only a ‘social’ treatment. Maybe it is ‘reproductive environment complexity’? If the authors disentangle this issue, then they can try to improve the title. I would see more precise (and more catching) something like “Early-life experience with male mating tactics shapes….”.
Changed title and adjusted social environment to sexual-social environment throughout the manuscript.
Methods
L103 Very nice number of replicates!
Thank you!
L125 Maybe habituation cylinder?
done
L131 How long was the real exposure to the male behaviour? I assume that until the females were sexually mature, the males were not courting them.
Experimental females were exposed to male behavior during their juvenile period (through observation of interactions between adult model females and males) and as young adults with direct experience. We addressed the role of both social learning and learning from direct experience in lines 171-174.
L158 Can 15-18 months can still be considered ‘early-life experience’ in this species? In many other poecilids it is close to the typical life span in the nature.
We’ve changed ‘early-life’ to “developmental” in title and throughout the manuscript.
L200 maybe about can be removed?
Removed.
L247 I would probably switch the order of the two tests, presenting first the statistical methods of the coercion experiment, in line with methods and results section.
Good idea. We’ve switched the order.
L254 If I understand correctly, a decrease in the time to solve the task can be also due to other factors. For instance, the fish could be more and more habituated to the apparatus and progressively increase the swimming speed. If in this experiment it is not possible to exclude these effects, please do not referee to changes in time as learning.
This is a very good point. To ascertain if tank habituation was the underlying mechanism explaining the pattern in our data, we examined whether or not species were engaging with the task faster on each successive trial, and found that they did not. We now include this analysis in our Results (Lines 331-337) and Discussion (lines 490-493).
Results
L286 Was the eigen value greater than 1? I don’t expect so. In case it is below 1, an analysis on the raw trait data may be more informative because the PC has probably lost a lot of behavioural variance.
Eigenvalue is 3.02 for PC1
L313 That is what I meant above. Probably the time measure is affected by factors other than learning. Maybe the time and the motivation better represent the approach to the task, or the style to tackle the task, rather than the learning performance. Please consider changing the text accordingly, I think it is quite important do don’t misinterpret this time effect. The main effect due to the route learning and the main result of the study should not be altered by this change.
We agree that differences in motivation to approach the task are very important to highlight, and are important to avoid misinterpretation. We therefore included both measures of solve time: the raw solve time that includes variation in motivation as well as the one that excludes the variation across individuals in how motivated they were to initiate engagement in the task (our motivation-controlled solve time). Importantly, we emphasize the accuracy differences across the treatment groups (which tell the same story as our differences in solve time). These accuracy measurements are independent of time and reveal statistically robust differences across treatments. By reporting all the measures (solve time, motivation-controlled solve time, and accuracy) we comprehensively describe the variation across all the treatments. We also now include a paragraph discussing some of the behavioral differences across these social rearing treatments that might influence how they approach this task (e.g. boldness and learning; see lines 550-572).
Discussion
L405 Can it be that there are other factors involved in the treatment? Maybe the males with a specific mating tactic have also other behavioural features. See for instance:
Kelley, J. L., Phillips, S. C., & Evans, J. P. (2013). Individual consistency in exploratory behaviour and mating tactics in male guppies. Naturwissenschaften, 100, 965-974.
Thank you for bringing up this point. This inspired us to discuss the behavioral differences that emerged from different sexual-social rearing treatments that may have contributed to the variation in learning performance we observed (again, see lines 550-572).
L471 Probably beside the paper suggested in introduction, these two can help expand the role of social environment in the present experiment (although they are not studies in fish):
Ashton, B. J., Ridley, A. R., Edwards, E. K., & Thornton, A. (2018). Cognitive performance is linked to group size and affects fitness in Australian magpies. Nature, 554(7692), 364-367.
Johnson-Ulrich, L., & Holekamp, K. E. (2020). Group size and social rank predict inhibitory control in spotted hyaenas. Animal Behaviour, 160, 157-168.
Thank you for these suggestions, we now include them in our Discussion (See line 504).
L491 I would check if this idea aligns with our most recent paper. It is a test on the effect of predictability on various cognitive functions in a poecilid.
Lucon-Xiccato, T., Montalbano, G., & Bertolucci, C. (2023). Adaptive phenotypic plasticity induces individual variability along a cognitive trade-off. Proceedings of the Royal Society B, 290(2001), 20230350.
This is a fascinating study and highly relevant to our findings (now cited in lines 523-526 and 543). Thank you for alerting us to this.
L506 I think that domain specificity was demonstrated earlier here (at least in fish):
Montalbano, G., Bertolucci, C., & Lucon-Xiccato, T. (2022). Cognitive phenotypic plasticity: environmental enrichment affects learning but not executive functions in a teleost fish, Poecilia reticulata. Biology, 11(1), 64.
My compliments for this interesting study!
Thank you so much for your kind and insightful comments.
Reviewer 2 Report
Comments and Suggestions for Authors
This manuscript is aimed to assess how early-life experiences with different males shapes female spatial cognition and harassment evasion ability. The authors raised females of Xiphophorus n. in different environments that varied by male mating strategies. Later they evaluated the abilities of the females of the different groups to solve a maze and in their responses to coercive males. The authors claim that the different experiences with courtship and coercion in the early rearing environment shape coercion evasion behaviors and spatial cognition in swordtails female. Specifically, they conclude that early life experiences in simpler social environments can lead to a better spatial behavior and that greater experience with coercion conducts to an improvement in harassment avoidance.
The study is well conducted, and the objectives are relevant.
-The authors say that the multiple sexual male strategies of this species create complex neural and cognitive challenges to females (line 75). It would be valuable that the authors discuss haw the results of this study can contribute to improve our understanding on the neural bases of cognition in fish.
- The unexpected results with the spatial cognition test may be due to the very low number of training trials in the maze. The authors should consider that the acquisition of an implicit habit of route learning likely need a greater number of trials. Increasing the number of trials would increase the level of accuracy at least in the learner groups. This possibility must be taken in account.
Minor comments
Fig 1G. I recommend marking the correct and incorrect doors of the maze.
Lines 326 and 327, note that some text is missing here.
Line 353, some words area repeated.
Author Response
This manuscript is aimed to assess how early-life experiences with different males shapes female spatial cognition and harassment evasion ability. The authors raised females of Xiphophorus n. in different environments that varied by male mating strategies. Later they evaluated the abilities of the females of the different groups to solve a maze and in their responses to coercive males. The authors claim that the different experiences with courtship and coercion in the early rearing environment shape coercion evasion behaviors and spatial cognition in swordtails female. Specifically, they conclude that early life experiences in simpler social environments can lead to a better spatial behavior and that greater experience with coercion conducts to an improvement in harassment avoidance.
The study is well conducted, and the objectives are relevant.
Thank you very much for your constructive feedback.
-The authors say that the multiple sexual male strategies of this species create complex neural and cognitive challenges to females (line 75). It would be valuable that the authors discuss haw the results of this study can contribute to improve our understanding on the neural bases of cognition in fish.
Very good point. We now address how our results add to our understanding of neural control of spatial cognition and coercion evasion behavior in the Discussion (see lines 600-611).
- The unexpected results with the spatial cognition test may be due to the very low number of training trials in the maze. The authors should consider that the acquisition of an implicit habit of route learning likely need a greater number of trials. Increasing the number of trials would increase the level of accuracy at least in the learner groups. This possibility must be taken in account.
Another good point. We include this possibility in our Discussion (see lines 496-498).
Minor comments
Fig 1G. I recommend marking the correct and incorrect doors of the maze.
Good suggestion. We’ve marked the correct and incorrect doors for the first barrier of the maze on Figure 1.
Lines 326 and 327, note that some text is missing here.
Thank you- this is corrected.
Line 353, some words area repeated.
Corrected—Thank you!